# Fused Triazinobenzimidazoles Bearing Heterocyclic Moiety: Synthesis, Structure Investigations, and In Silico and In Vitro Biological Activity

**DOI:** 10.3390/molecules28135034

**Published:** 2023-06-27

**Authors:** Kameliya Anichina, Nikolai Georgiev, Nikolay Lumov, Dimitar Vuchev, Galya Popova-Daskalova, Georgi Momekov, Emiliya Cherneva, Rositsa Mihaylova, Anelia Mavrova, Stela Atanasova-Vladimirova, Iskra Piroeva, Denitsa Yancheva

**Affiliations:** 1Department of Organic Synthesis, University of Chemical Technology and Metallurgy, 8 Kliment Ohridski Blvd., 1756 Sofia, Bulgaria; anmav@abv.bg; 2Institute of Organic Chemistry with Centre of Phytochemistry, Bulgarian Academy of Sciences, Acad. G. Bonchev Str. Bl. 9, 1113 Sofia, Bulgaria; nikolay.lumov@orgchm.bas.bg (N.L.); cherneva@pharmfac.mu-sofia.bg (E.C.); denitsa.pantaleeva@orgchm.bas.bg (D.Y.); 3Department of Infectious Diseases, Parasitology and Tropical Medicine, Medical University, 15A Vasil Aprilov Blvd., 4002 Plovdiv, Bulgaria; vutchev_d@abv.bg (D.V.); galia_val@mail.bg (G.P.-D.); 4Faculty of Pharmacy, Medical University of Sofia, 2 Dunav Str., 1000 Sofia, Bulgaria; gmomekov@pharmfac.mu-sofia.bg (G.M.); rmihaylova@pharmfac.mu-sofia.bg (R.M.); 5Institute of Physical Chemistry, Bulgarian Academy of Sciences, Build. 11, 1113 Sofia, Bulgaria; statanasova@ipc.bas.bg (S.A.-V.); piroeva@ipc.bas.bg (I.P.)

**Keywords:** benzimidazole, 1,3,5-thiazine, DFT, *Trichinella spiralis*, antinematodal activity, anticancer, drug-likeness

## Abstract

[1,3,5]Triazino[1,2-*a*]benzimidazole-2-amines bearing heterocyclic moiety in 4-position were synthesized. The compounds were characterized by elemental analysis, IR, ^1^H-NMR, ^13^C-NMR, and HRMS spectroscopy. The molecular geometry and electron structure of these molecules were theoretically studied using density functional theory (DFT) methods. The molecular structure of the synthesized fused triazinobenzimidazole was confirmed to correspond to the 3,4-dihydrotriazinobenzimidazole structure through the analysis of spectroscopic NMR data and DFT calculations. The antinematodic activity was evaluated in vitro on isolated encapsulated muscle larvae (ML) of *Trichinella spiralis*. The results showed that the tested triazinobenzimidazoles exhibit significantly higher efficiency than the conventional drug used to treat trichinosis, albendazole, at a concentration of 50 μg/mL. The compound **3c** substituted with a thiophen-2-yl moiety exhibited the highest anthelmintic activity, with a larvicidal effect of 58.41% at a concentration of 50 μg/mL after 24 h of incubation. Following closely behind, the pyrrole analog **3f** demonstrated 49.90% effectiveness at the same concentration. The preliminary structure-anti-*T. spiralis* activity relationship (SAR) of the analogues in the series was discussed. The cytotoxicity of the benzimidazole derivatives against two normal fibroblast cells (3T3 and CCL-1) and two cancer human cell lines (MCF-7 breast cancer cells and chronic myeloid leukemia cells AR-230) was evaluated using the MTT-dye reduction assay. The screening results indicated that the compounds showed no cytotoxicity against the tested cell lines. An in silico study of the physicochemical and pharmacokinetic characteristics of the novel synthesized fused triazinobenzimidazoles showed that they were characterized by a significant degree of drug-likeness and optimal properties for anthelmintic agents.

## 1. Introduction

Trichinosis is a zoonotic helminthic infection caused by roundworms from the genus *Trichinella* that occurs with fever, edema, high eosinophilia, and toxoallergic reactions. Humans become infected after consuming meat products contaminated with *Trichinella* larvae [1]. For 2020, nine countries in the European Union reported 181 cases of trichinellosis, of which 117 were confirmed. Bulgaria, Italy, and Poland accounted for 88% of all confirmed cases reported in 2020 [2].

The main drugs used for treating trichinosis, albendazole and mebendazole, belong to the benzimidazole anthelmintics group (Figure 1). However, these medications have limited effectiveness due to their inability to effectively combat encapsulated *Trichinella* larvae in muscles. Additionally, their bioavailability is restricted, and resistance to these drugs has been observed, similar to the first benzimidazole anthelmintic, thiabendazole, whose clinical use today has been significantly reduced [3,4]. Therefore, there is a pressing need to develop new anti-*Trichinella* drugs that are more potent, less toxic, and devoid of adverse effects.

It is known that the benzimidazole heterocycle is present in the structure of many trichinellosis agents [5,6]. In previous studies, we explored the synthesis and anthelmintic activity of some 2-arylidene-substituted thiazolo[3,2-a]benzimidazoles (Figure 1). These compounds exhibited high efficacy against *Trichinella spiralis (T. spiralis)* both in vitro and in vivo [7]. Moreover, certain bis(benzimidazol-2-yl)amines showed higher in vitro activity against isolated muscle larvae of *T. spiralis* than albendazole. The in vivo screening of the intestinal phase of *T. spiralis* revealed 100% effectiveness of the compounds at oral dosages of 50 and 100 mg/kg mw, while albendazole possesses 100% efficacy only at a dose of 100 mg/kg mw [8]. Furthermore, we investigated the antiparasitic properties of piperazine derivatives of 5(6)-substituted-(1*H*-benzimidazol-2-yl-thio)acetic acids. The most active compounds exhibited efficacy ranging from 96.0% to 100% in vitro [9]. In our research, we also synthesized a small library of benzimidazolyl-2-hydrazones, incorporating different substituted aromatic rings or a 1,3-benzodioxole aromatic moiety (Figure 1). These compounds surpassed the positive controls, albendazole and ivermectin, in terms of their in vitro activity against *Trichinella* larvae [10].

On the other hand, triazines are another class of heterocyclic compounds that have demonstrated significant potential as antiparasitic agents; more especially, the 1,3,5-triazine derivatives have shown excellent anti-malarial activity [11,12,13]. Agarwal et al. conducted a study where they synthesized a series of 2,4,6-trisubstituted-1,3,5-triazines (Figure 1) and evaluated their in vitro antimalarial activity against *P. falciparum*. The analogs containing N-methyl piperazine or imidazole moieties displayed a minimum inhibitory concentration (MIC) of 1 µg/mL [14]. Similarly, Bhat et al. reported on a new generation of 7-chloro-4-aminoquinoliyl-1,3,5-triazines [15]. The hybrid molecule containing piperidine at 1,3,5-triazine exhibited notable activity against malaria parasites [15].

In the treatment of trichinellosis, the latest trend involves combining albendazole with an antimalarial drug [16]. This combination approach is considered the most effective strategy for delaying drug resistance [17]. For instance, the combined albendazole-mefloquine low-dose regimen has shown great efficacy in reducing the *Trichinella* parasite burden and restoring normal tissue architecture. Furthermore, the synergistic activity between the two drugs helps mitigate their side effects and enhances their biological activity.

Given these developments, the design and synthesis of compounds that incorporate both the benzimidazole and 1,3,5-triazine pharmacophores in their molecular structure hold pharmacological interest for the development of novel anti-*Trichinella spiralis* agents. We anticipate that the results presented in this study will contribute to a more comprehensive understanding of the relationship between the structural properties and the anti-*Trichinella* activity of triazinobenzimidazoles. This, in turn, will allow the determination of further directions for modifying the target compounds in order to obtain new derivatives with more potent pharmacological action and lower toxicity.

## 2. Results and Discussion

### 2.1. Synthesis

Within the frame of current research, we have synthesized several hybrid molecules containing fused triazinobenzimidazole azaheterocycles and pyridine, furan, thiophene, indole, and pyrrole moiety according to the reaction shown in Figure 1. The interaction of 2-guanidino-benzimidazole **1** with the corresponding commercial heteroaromatic aldehyde **2a**–**f** was carried out in a medium of absolute ethanol at a molar ratio of the reactants of 1:1 by using piperidine as a catalyst [13,18,19]. This catalyzed cyclization methodology provides simple and mild reaction conditions, a short reaction time, and excellent yield. 

The guanidine group exhibits strong basic properties, making it highly reactive towards aldehydes, resulting in the formation of compounds similar to Schiff bases (intermediate 1, Figure 2A). A piperidine-catalyzed imination of 2-guanidinobenzimidazole with an aldehyde is also possible, as described in Figure 2B. In this case, the aldehyde and piperidine undergo condensation, yielding piperididine hydroxide. Following that, 2-guanidinobenzimidazole undergoes nucleophilic addition to the hydroxide salt, leading to the formation of intermediate **0** and the elimination of water (H_2_O). Subsequently, piperidine is expelled, resulting in the formation of intermediate **1** [20]. In the next step, piperidine cleaves the imide hydrogen atom (N-H) from the benzimidazole molecule, generating intermediate **2**. Intermediate **3** is subsequently protonated, resulting in the generation of intermediate **4**. In the final step of the reaction, it is anticipated that the hydrogen atom attached to N3 will be transferred to the N-H group at C2, leading to the formation of an amino group.

The structures of the synthesized 2-amino-4-heteryl-3,4-dihydro[1,3,5]triazino[1,2-a]benzimidazoles **3d**–**f** were elucidated with the help of IR, NMR, and HRMS (ESI) spectra as well as elemental analyses. The characteristic of these compounds is that they could exist in three tautomeric forms: 3,4-dihydro-(A), 1,4-dihydro-(B), and 4,10-dihydro-(C) (Figure 2). The spectral data, in particular the singlet at 6.80–7.06 ppm of H-4 in ^1^H NMR spectra, together with the signal of C-4 at 61.4–66.0 ppm in the ^13^C NMR spectra, confirmed the formation of the 3,4-dihydro form. These results match the data from the spectroscopy study of analogous compounds [13,18].

In addition, the chemical shifts for the three possible tautomer forms of compound **3b** were predicted by the GIAO method using B3LYP functional and 6-311++G** basis set calculations in DMSO solvent to determine the preferred tautomeric form. By comparing the calculated chemical shifts of the tautomeric forms, it is possible to identify the form that best matches the experimental spectral data. The predicted chemical shifts demonstrated that form A (3,4-dihydro) is characterized by the lowest value for the C-4 signal compared to forms B and C (Appendix A). According to the 2D spectrum (HSQC) (Appendix A), for compound **3b**, the signal of H-4 at 6.91 ppm in the ^1^H-NMR spectrum corresponds to the signal of C-4 at 64.7 ppm in the ^13^C NMR spectra, which also matches the data provided earlier by Dolzhenko et al. [18]. 

### 2.2. Computational Methods

The molecular geometry of the intermediates included in the suggested mechanism (Figure 2) was studied by DFT calculations at the B3LYP/6-311++G** level of theory in gas phase. 

The most probable molecular structure of this intermediate 2 (Figure 2) is illustrated by the 4-pyridyl derivative **3b** in Figure 3. In order to undergo subsequent intramolecular cyclization, the azomethine C-atom needs to be oriented towards the deprotonated nitrogen from the benzimidazole ring. According to the calculation, the molecular structure of intermediate **2** is stabilized with a 3.027 Å distance between the N^-^ and C-atoms (Figure 3). This molecular structure allows the formation of cyclic anionic intermediate **3**, where the newly formed C-4-N-5 bond (the numbering of the atoms is represented in Figure 2) has 1.487 Å length and the azomethine bond is replaced by a single N-3-C-4 bond of 1.421 Å length (Figure 3). Protonation of the cyclic anionic intermediate **3** yields intermediate 4, where the bond lengths of both N-3-C-4 and C-4-N-5 are equal and the heteroaromatic fragment at C4 (the 4-pyridyl fragment) gets substituent oriented in a perpendicular plane in regards to the triazinobenzimidazolic fragment (Figure 3) as it is in the final product of the reaction (see below **3b** in Figure 4). In the final step of the reaction, it is anticipated that the H attached to N-1 is transferred to the N-H group at C-2 and the amino group is formed.

The molecular structure and tautomerism of the newly synthesized compounds were studied by DFT B3LYP/6-311++G** calculations in water medium. According to quantum chemical calculations, the most stable tautomeric form in all compounds is the 3,4-dihydro form (Figure 4). 

The energy difference between the less stable forms—B (1,4-dihydro-) and C (4,10-dihydro-)—is in the range 8–15 kJ·mol^−1^, which implies that practically only the 3,4-dihydro form is present in water medium. The two less stable tautomers are characterized by a significantly smaller energy difference between them. 

The quantum chemical calculations showed that in compounds **3a**–**f,** the 3,4-dihydro [1,3,5]triazino[1,2-a]-benzimidazole fragment is planar, and the condensed heteroaromatic fragment lies almost perpendicularly with dihedral angle N3-C4-C1′-C2′ (or O′/N′ from the heterocycle) in the interval 50–80°. Having a small number of single bonds around which rotation can take place makes the molecules stable and allows them to easily stay in that configuration. In regard to the single bond between the triazine part and the condensed heterocycle, the cis-configuration (with the heteroatoms oriented below the dihydrotriazinobenzimidazole plane) is preferred over the trans-configuration.

### 2.3. Antihelmintic Activity

The synthesized compounds were evaluated for anthelmintic activity against isolated *T. spiralis* muscle larvae (ML) according to an established protocol [10,21]. The antiparasitic drugs albendazole and ivermectin were used as standard references. The obtained results, summarized in Table 1, made it possible to clearly infer that the triazinobenzimidazoles are found to be more active in comparison with the standards at both tested concentrations. The studied compounds reduced the viability of *T. spiralis* larvae in a time- and dose-dependent manner, analogically to other benzimidazoles [10].

The derivatives **3c** and **3f,** containing a five-membered ring with a heteroatom in a 4-position—thiophene and pyrrole, respectively—showed a remarkable larvacidal effect on *Trichinella* ML. The thiophene analog **3c** exhibited a 58.4% larvicidal effect at a concentration of 50 μg/mL after 24 h and killed the total larvae (100% effectiveness) at a concentration of 100 μg/mL after 48 h of incubation. The replacement of the sulfur atom of compound **3c** with the NH resulted in compound **3f** being endowed with anti-*Trichinella spiralis* activity, comparable to that of the anthelmintic ivermectin at 50 μg/mL. However, with increasing concentration and incubation time, the activity of **3f** is higher than that of the macrocyclic lactone ivermectin. The fused triazinobenzimidazole bearing benzo[1,3]dioxole moiety **3d** has greater lethal activity against ML in comparison to the indole derivate **3e**. For example, the larvicidal effects of **3d** were 42.1% and 52.4%, respectively, after 24 h and 48 h at a concentration of 50 μg/mL which were better than those of compounds **3e** (31.5% and 36.9%). These results correspond with an earlier study conducted by us on the anti-trichinellosis activity of some thiazolo[3,2-a]benzimidazolones, according to which the derivatives containing benzo[1,3]dioxole cores are more active than their indole analogs [7]. The benzo[1,3]dioxole heterocycle is also present in the structure of the benzophenanthridine alkaloid sanguinarine, which had a lethal effect on newborn larvae, muscle larvae, and adults of T. spiralis in vitro and caused 100% destruction of the muscle larvae in vitro at a concentration of 30 mg/mL [22]. 

The least active antiparasitic compounds of the series, those with pyridine substituents at positions 4 (**3a** and **3b**), exhibit threefold higher in virto activity than the anthelmintic albendazole, traditionally used to treat trichinosis infection [23,24]. Furthermore, the position of the nitrogen atom in the pyridine heterocycle has an effect on the larvicidal effects of **3a** and **3b**. For example, the efficacy of the pyridine-2-yl derivative **3a** against the parasitic larvae was respectively 37.2, 48.8, and 72.8% (at 50 μg/mL), which was higher than that of the compound **3b** (21.3, 47.5, and 65.5%, at the same concentration after 24 h, 48 h, and 72 h incubation periods at 37 °C).

### 2.4. Scanning Electron Microscopy

The dead larvae treated with 2-amino-4-(thiophen-2-yl)-3,4-dihydro[1,3,5]triazino[1,2-a]benzimidazole 3c at a concentration of 50 μg/mL for 48 h were isolated from the solution and dehydrated as described in [25]. Prepared larvae were fixed on a microscope slide coated with a thin gold film (<10 nm). Gold at that thickness will have no effect on *T. spiralis* ML. After sputter coating with gold, they were observed in a JSM 6390 electron microscope (JEOL, Tokyo, Japan) in regimes of SEI. The accelerating voltage was 5 kV. The results of the scanning electron microscope examination are shown in Figure 5.

In the current study, the electron microscopy scans showed destruction of the normal structure of the cuticle of *T. spiralis* ML [26,27], expressed in the presence of areas with fissures and smoothing of the annulations (Figure 5A). Figure 5B shows that the larva’s cuticle was severely damaged and destructured. These observations indicate the antinematodal effectiveness of the newly synthesized thiazinobenzimidazoles because the cuticle is the main route of drugs’ passage into the nematodes, with subsequent destruction of the larvae’s surface [26].

### 2.5. In Vitro Evaluation of Cytotoxicity

Potential antiparasitic drug candidates must not only possess activity against parasites but also not exhibit toxic effects on the host organism [28,29].

In connection with this, the preliminary cytotoxicity of the 2-amino-3,4-dihydro-[1,3,5]triazino[1,2-*a*]benzimidazole derivatives **3a**–**f**, which demonstrated significant anthelmintic activity against *T. spiralis* larvae, was evaluated by determination of the cellular viability in two normal fibroblast cells—3T3 (mouse embryo fibroblasts) and CCL-1 (murine fibroblasts)—and in two cancer human cell lines of different histological origin: MCF-7 (ER-positive breast adenocarcinoma) and AR-230 (BCR-ABL-positive chronic myeloid leukemia). 

Cellular viability was estimated after a 72-h incubation period in a concentration range of 1 to 250 µg·mL^−1^ using the methylthiazolyldiphenyltetrazolium bromide (MTT) reduction assay.

The newly synthesized compounds **3a**–**f** did not show cytotoxicity against the cell lines in the screened in vitro models. Their half-inhibitory concentrations (IC_50_) in tested cell lines are >100 μM, analogically to earlier published data by Hranjec et al. [19] for the antiproliferative activity of 2-amino-4-heteryl-4,10-dihydro[1,3,5]triazino[1,2-*a*]benzimidazoles. These results show that the anthelmintic activity of the tested triazinobenzimidazoles is not due to general toxicity but can be attributed to their selective action against parasitic larvae. However, for further studies, additional examination of the toxicology of the most active antiparasitic agents needs to be performed.

### 2.6. Prediction of the Physico-Chemical Properties and Drug-Likeness

The fused thiazolobenzimidazoles **3a**–**f** were subjected to computational ADME, pharmacokinetic, and drug-likeness evaluation using the web tool SwissADME [30].

Data from the SwissADME assessment of drug-likeness (Table 2) indicated that all compounds obtained obey Lipinski’s rule. Containing less than 5 hydrogen bond donors and less than 10 hydrogen bond acceptors, a molecular weight less than 500, and a logP less than 5, these products possess physicochemical properties that give reason to expect very good bioavailability. 

The target molecules are of low molecular weight, indicating good oral absorption. Based on the calculated consensus logP values, obtained as the average of the five available logP methods [30], synthesized compounds proved to be slightly lipophilic. Components containing larger heterocyclic substituents, as well as those containing oxygen and sulfur as heteroatoms, showed higher logP values. 

The predicted TPSA values of the study derivatives are in the range 81.12–97.47 Å2, indicating good intestinal absorption. The low number of rotatable bonds in the triazinobenzimidazoles corresponds to sufficient oral bioavailability. They are soluble or moderately soluble in water, according to the SwissADME server [30]. Solubility data for the compounds indicate that they are sufficiently soluble and can be used as oral drug candidates [31,32].

Considering the relative properties of the different tautomeric structures, the 3,4-dihydro- and 1,4-dihydrotriazinobenzimidazole tautomeric forms of the studied compounds are characterized by lower polar surface (TPSA) and partition coefficients (milogP) than the third tautomeric form—the 4,10-dihydrotriazinobenzimidazole form, as can be seen in Appendix A.

## 3. Materials and Methods

### 3.1. General Procedures

Commercially available reagents (2-guanidinobenzimidazole, 95%), and piperonal, 99%, were obtained from Sigma-Aldrich (Steinheim, Germany). The other heterocyclic aldehydes (pyridine-2-carboxaldehyde, 99%; pyridine-4-carboxaldehyde, 97%; thiophene-2-carboxaldehyde, 98+%; indole-3-carboxaldehyde, 99%; and pyrrole-2-carboxaldehyde, 99%) were purchased from Alfa-Aesar (Heysham, UK). All commercially available solvents were obtained from Alfa Aesar (Heysham, UK) and were used without purification or drying. Reactions were routinely monitored by thin layer chromatography (TLC) on pre-coated silica gel plates, ALUGRAM SIL G/UV254, 0.20 mm thick (Macherey-Nagel, Dueren, Germany), eluted by a benzene-methanol mixture (3:1, *v*/*v*), and visualized using UV light. The melting points (mp) were measured using an Electrothermal AZ 9000 3MK4 apparatus (Stone, Staffordshire, UK) and were uncorrected. The IR spectra of the synthesized compounds in a solid state were recorded on a Bruker Tensor 27 FT spectrometer (Billerica, MA, USA) in ATR (attenuated total reflectance) mode with a diamond crystal accessory. The spectra were referenced to air as a background by accumulating 64 scans at a resolution of 2 cm**^−^**^1^. The ^1^H and ^13^C NMR spectra were recorded on a Bruker Avance 600 MHz spectrometer at room temperature (303 K) using DMSO-*d*_6_ as a solvent. Chemical shifts (δ) are reported in parts per million (ppm). Coupling constants J are given in Hertz (Hz), and spin multiplicities are given as singlet (s), broad singlet (br s), doublet (d), doublet of doublets (dd), triplet (t), triplet of doublets (td), and multiplet (m). Mass spectrometric analyses were carried out on a Q Exactive Plus Mass Spectrometer (ThermoFisher Scientific, Waltham, MA, USA) equipped with a heated electrospray ionization (HESI-II) probe (ThermoFisher Scientific). Operating conditions for the HESI source used in a positive ionization mode were: spray voltage +3.5 kV, 320 °C capillary and probe heater temperatures, sheath gas flow rate 36 a.u., auxiliary gas flow rate 11 a.u., and S-Lens RF level 50.00. Full MS—SIM was used as an MS experiment in negative and positive modes, where resolution, automatic gain control (AGC) target, maximum injection time (IT), and mass range were 70,000 (at *m*/*z* 200), 3e6, 100 ms, and *m*/*z* 100–500, respectively. Xcalibur (Thermo Fisher Scientific, Waltham, MA, USA) ver. 4.0 was used for data acquisition and processing. The microanalyses for C, H, N, and S were performed on a PerkineElmer elemental analyzer (Waltham, MA, USA). SEM analyses were performed on a JSM 6390 electron microscope (JEOL, Tokyo, Japan) in regimes of secondary electron image (SEI). The SEI images give information about the investigated worms. The accelerating voltage was 5 kV. Prepared worms were fixed on a microscope slide coated with a thin gold film (<10 nm). Gold at that thickness will have no effect on worms.

### 3.2. Synthesis

#### 3.2.1. General Procedure for the Synthesis of Compound **3a**–**f**

A 2-guanidinobenzimidazole **1** (1 g, 5.70 mmol), a heteroaromatic aldehyde **2a**–**f** (5.70 mmol), and 0.2 mL piperidine (2.00 mmol) in absolute ethanol (13–15 mL) were heated under reflux for 2-4 h until precipitation took place. After cooling to room temperature, the precipitate formed **3a**–**f** was filtered, washed with ethanol, dried, and further purified.

#### 3.2.2. Compound Data

##### 2-Amino-4-(pyridine-2-yl)-3,4-dihydro[1,3,5]triazino[1,2-a]benzimidazole (**3a**)

White powder (1.010 g, 67%), reaction time 4 h, mp > 250 °C, Rf = 0.516; IR (KBr) υ/cm^−1^: 3295, 3250, 1680, 1636, 1592, 1515, 1462, 739, 761. ^1^H NMR (600 MHz, DMSO-*d*_6_): 8.55 (d, 1H, *J* = 4.63 Hz, 3′-H), 8.21 (br s, 1H, NH), 7.82 (td, 1H, *J* = 7.8; 2.23 Hz), 7.36 (dt, 1H, *J* = 6.2; 0.8 Hz), 7.29 (d, 1H, *J* = 7.8 Hz), 7.23 (d, 1H, *J* = 7.9 Hz), 6.93 (t, 1H, *J* = 7.6; 1.1 Hz), 6.90 (d, 1H, *J* = 7.7 Hz), 6.81 (td, 1H, *J* = 7.8; 0.4 Hz); 6.81 (s, 1H, 4-H); 6.53 (br s, 2H, NH_2_). ^13^C NMR (150 MHz, DMSO-*d*_6_) δ (ppm): 158.6, 150.2, 143.4, 138.1, 131.7, 124.8, 121.4, 121.1, 119.6, 108.5, 67.1, 56.5. HRMS (ESI) *m*/*z*: Calcd. for C_14_H_12_N_6_: 264.1123; Found: 265.1193 [M + H]^+^. Anal. Calcd. for C_14_H_12_N_6_: C, 63.62; H, 4.58; N, 31.80; Found: C, 62.91; H, 4.50; N, 30.93.

##### 2-Amino-4-(pyridine-4-yl)-3,4-dihydro[1,3,5]triazino[1,2-a]benzimidazole (**3b**)

White powder (1.065 g, 71%), reaction time 4 h, mp 208–210 °C, Rf = 0.443; IR (KBr) υ/cm^−1^: 3381, 3235, 1668, 1622, 1602, 1537, 1515, 740, 852. ^1^H NMR (600 MHz, DMSO-*d*_6_): 8.58 (d, 2H, *J* = 4.1 Hz), 8.25 (br s, 1H, NH), 7.27–7.26 (s, 3H), 6.97 (t, 1H, *J* = 7.5 Hz), 6.91 (s, 1H), 6.85–6.84 (m, 2H), 6.60 (br s, 2H, NH_2_). ^13^C NMR (150 MHz, DMSO-*d*_6_) δ (ppm): 155.7, 150.9, 149.0, 131.4, 121.7, 121.3, 119.8, 116.6, 108.5, 64.7. HRMS (ESI) *m*/*z*: Calcd. for C_14_H_12_N_6_: 264.1123; Found: 265.1193 [M + H]^+^. Anal. Calcd. for C_14_H_12_N_6_: C, 63.62; H, 4.58; N, 31.80; Found: C, 63.58; H, 4.53; N, 31.78.

##### 2-Amino-4-(thiophen-2-yl)-3,4-dihydro[1,3,5]triazino[1,2-a]benzimidazole (**3c**)

White powder (1.353 g, 88%), reaction time 4 h, mp > 250 °C, Rf = 0.584; IR (KBr) υ/cm^−1^: 3429, 3320, 3232, 1621, 1636, 1592, 1515, 1462, 739, 761. ^1^H NMR (600 MHz, DMSO-*d*_6_): 8.23 (br s, 1H, NH), 7.50 (d, 1H, *J* = 4.9 Hz), 7.30 (d, 1H, *J* = 2.8 Hz), 7.22 (d, 1H, *J* = 7.6 Hz), 7.23 (s, 1H), 6.99 (td, 1H, *J* = 3.5; 1.5 Hz), 6.96 (t, 1H, *J* = 7.5 Hz), 6.85 (t, 1H, *J* = 7.5 Hz); 6.53 (br s, 2H, NH_2_). ^13^C NMR (150 MHz, DMSO-*d*_6_) δ (ppm): 144.5, 143.6, 131.5, 127.8, 127.0, 126.9, 121.5, 119.6, 116.5, 108.7, 62.0. HRMS (ESI) *m*/*z*: Calcd. for C_13_H_11_N_5_S: 269.0735; Found: 270.0805 [M + H]^+^. Anal. Calcd. for C_13_H_11_N_5_S: C, 57.98; H, 4.12; N, 26.00; Found: C, 57.90; H, 3.98; N, 25.87.

##### 2-Amino-4-(benzo[1,3]dioxol-5-yl)-3,4-dihydro[1,3,5]triazino[1,2-a]benzimidazole (**3d**)

White powder (1.140 g, 65%), reaction time 2 h, mp 215–218 °C, Rf = 0.640; IR (KBr) υ/cm^−1^: 3346, 3237, 2970, 1643, 1651, 1453, 1244, 1173, 745, 762. ^1^H NMR (600 MHz, DMSO-*d*_6_): 8.00 (br s, 1H, NH), 7.22 (d, 1H, *J* = 7.8 Hz), 6.94–6.88 (m, 3H), 6.83–6.76 (m, 3H), 6.66 (s, 1H), 6.44 (br s, 2H, NH_2_), 6.00 (d, 2H, *J* = 12.6). ^13^C NMR (150 MHz, DMSO-*d*_6_) δ (ppm): 154.0, 148.3, 148.2, 143.9, 135.0, 131.7, 121.2, 120.5, 119.4, 116.4, 108.8, 106.7, 101.8, 66.0. HRMS (ESI) *m*/*z*: Calcd. for C_16_H_13_N_5_O: 307.1069; Found: 308.1137 [M + H]^+^. Anal. Calcd. for C_16_H_13_N_5_O_2_: C, 62.53; H, 4.26; N, 22.79; Found: C, 63.05; H, 4.38; N, 22.60.

##### 2-Amino-4-(1H-indol-3-yl)-3,4-dihydro[1,3,5]triazino[1,2-a]benzimidazole (**3e**)

White powder (1.536 g, 89%), reaction time 3 h, mp > 250 °C, Rf = 0.487. ^1^H NMR (600 MHz, DMSO-*d*_6_): 11.26 (d, 1H, *J* = 1.4 Hz, NH), 7.90 (br s, 1H, NH), 7.72 (d, 1H, *J* = 2.6 Hz), 7.37 (t, 2H, *J* = 7.2 Hz), 7.17 (d, 1H, *J* = 7.8 Hz), 7.06 (t, 1H, *J* = 7.9 Hz); 7.01 (s, 1H); 6.93–6.85 (m, 2H), 6.80 (d, 1H, *J* = 7.4 Hz), 6.72 (t, 1H, *J* = 7.4 Hz), 6.33 (s, 2H, NH_2_). ^13^C NMR (150 MHz, DMSO-*d*_6_) δ (ppm): 156.2, 154.2, 143.7, 137.3, 132.0, 125.7, 124.5, 122.2, 120.9, 119.8, 119.4, 119.0, 116.1, 114.8, 112.4, 61.7. HRMS (ESI) *m*/*z*: Calcd. for C_17_H_14_N_6_: 302.1280; Found: 303.1350 [M + H]^+^. Anal. Calcd. For C_17_H_14_N_6_: C, 67.54; H, 4.67; N, 27.80; Found: C, 68.05; H, 4.58; N, 27.60.

##### 2-Amino-4-(1H-pyrrol-2-yl)-3,4-dihydro[1,3,5]triazino[1,2]benzimidazole (**3f**)

White powder (1.238 g, 86%), reaction time 3 h, mp > 250 °C, Rf = 0.542. ^1^H NMR (600 MHz, DMSO-*d*_6_): 11.29 (s, 1H, NH), 7.91 (br s, 1H, NH), 7.18 (d, 1H, *J* = 7.8 Hz), 6.91 (t, 1H, *J* = 7.2 Hz), 6.80 (d, 1H, *J* = 1.6 Hz), 6.74 (t, 1H, *J* = 7.5 Hz); 6.61 (s, 1H); 6.41–6.29 (m, 2H+ 2H, NH_2_), 6.06 (q, 1H, *J* = 6.3, 2.6 Hz). ^13^C NMR (150 MHz, DMSO-*d*_6_) δ (ppm): 156.5, 154.3, 143.8, 132.2, 129.0, 121.1, 120.2, 119.2, 116.2, 108.8, 107.9, 61.4. HRMS (ESI) *m*/*z*: Calcd. for C_13_H_12_N_6_: 252.1123; Found: 253.1194 [M + H]^+^. Anal. Calcd. For C_13_H_12_N_6_: C, 61.89; H, 4.79; N, 33.31; Found: C, 61.55; H 4.82; N, 33.04.

### 3.3. Computational Methods

The molecular geometry of the possible conformers and tautomeric forms was studied by quantum-chemical methods using the Gaussian 09 package of programs [33]. All structures were optimized within density functional theory (DFT) [34], employing the hybrid B3LYP functional in conjunction with the 6-311++G** basis set in a water medium [35,36]. The absence of imaginary frequencies was used to confirm that all optimized structures correspond to true minima on the potential energy surface.

### 3.4. Anthelmintic Study

The parasitological experiments in vitro for anti-*T. spiralis* activity of the tested compounds **3a**–**f** were performed as previously described [10,21].

*Trichinella spiralis*-encapsulated ML were obtained from the National Centre for Infectious and Parasitic Diseases, Department of Parasitology and Tropical Medicine, Sofia, Bulgaria. 

For the assay, 100 larvae in 1 mL of physiological solution were treated with 50 and 100 μg·mL^−1^ concentrations of experimental compounds dissolved in dimethyl sulfoxide (DMSO). Albendazole and ivermectin were used in this test as positive controls, and a sample with DMSO (1 mL) served as a negative control. After the incubation at 37 °C, the viability of the parasites was determined by direct microscopy at 24 h, 48 h, and 72 h, respectively. The efficacy percentage of newly synthesized benzimidazole hydrazones was calculated as the average value of three experiments [10].

### 3.5. Cytotoxicity Study

#### 3.5.1. Preparation of Cell Cultures

The in vitro cytotoxicity of the compounds was assessed in a panel of cell lines of different origin and characteristics, namely the normal murine fibroblast cell lines 3T3 (mouse embryo fibroblasts) and CCL-1 (fibroblast adipocytes), and two malignant human cell lines MCF-7 (hormone-dependent breast adenocarcinoma) and AR-230 (BCR-ABL positive chronic myeloid leukemia). All cell lines were purchased from the German Collection of Microorganisms and Cell Cultures (DSMZ GmbH, Braunschweig, Germany). Cell cultures were cultivated in a growth medium RPMI 1640 supplemented with 10% fetal bovine serum (FBS), 5% L-glutamine and incubated under standard conditions of 37 °C and 5% humidified CO_2_ atmosphere. The 3T3 cells were cultivated in Dulbecco’s Modified Eagle Medium (DMEM), containing 10% FCS, with supplements of L-glutamine, sodium pyruvate, antibiotics, and non-essential amino acids, and allowed to adhere overnight in a CO_2_ incubator at 37 °C. The cells were kept in log phase by supplementing with fresh media after removing cell suspension aliquots two or three times each week.

#### 3.5.2. Cell Viability Assay

The cytotoxicity of 2-amino-3,4-dihydro-[1,3,5]triazino[1,2-*a*]benzimidazole derivatives **3a**–**f** was determined using Mossman’s [37]. MTT [3-(4,5-dimethylthiazol-2-yl)-2,5-diphenyltetrazolium bromide] assay. The analysis is based on the reduction of the yellow tetrazolium compound MTT to water-insoluble violet formazan crystals. The cells were treated with different concentrations of compounds **3a**–**f** (1, 5, 10, 12.5, 25, 50, 75, 100, 125, and 250 µg·mL^−1^) in a final volume of 100 μL/well in triplicate wells for each treatment for 72 h at 37  °C in a 5% CO_2_ incubator. After the exposure period, MTT solution (10 mg·mL^−1^) aliquots (100 µL/well) were added to each well. The plates were further incubated for 4 h at 37 °C, and the MTT-formazan crystals formed were dissolved through the addition of 110 mL of 5% HCOOH in 2-propanol, and the absorption was measured at 580 nm using a microplate reader (Labexim LMR-1, Lengau, Austria). All studies were repeated at least twice.

## 4. Conclusions

A series of new 4-heteryl-2-amino-3,4-dihydro[1,3,5]triazino[1,2-*a*]benzimidazole derivatives were synthesized via base-catalyzed cyclization methodology. This approach offered several advantages, including simple and mild reaction conditions, a short reaction time, and excellent yields. The molecular structure of the obtained fused triazinobenzimidazole-heteroaromitic compounds, containing pyridine, thiophene, indole, and pyrrole substituents, was confirmed to correspond to the 3,4-dihydrotriazinobenzimidazole structure based on spectroscopic NMR data as well as DFT calculations. The anthelmintic activity study against isolated *T. spiralis* muscle larvae (ML) revealed that the newly synthesized 4-heteryl-2-amino-3,4-dihydro[1,3,5]triazino[1,2-a]benzimidazoles exhibit mild to excellent larvicidal effects in a time- and dose-dependent manner. The thiophene analog **3c** showed remarkable efficacy—58.4% larvicidal effect at a concentration of 50 μg/mL after 24 h and 100% effectiveness at a concentration of 100 μg/mL after 48 h of incubation. The substituent’s nature showed a clear effect on the larvicidal properties: concerning the five-membered rings, the thiophene substitution provided a more potent larvicidal effect than pyrrole; within the six-membered rings, the 2-pyridine proved more effective than the 4-analog; and the benzo[1,3]dioxole core proved more effective than the indole core. The cytotoxicity testing on two normal fibroblast cells, 3T3 and CCL-1, and on two cancer human cell lines of different histological origin, MCF-7 and AR-230, proved that all newly synthesized compounds are non-toxic. The SwissADME assessment of the drug-likeness predicted very good bioavailability of the compounds. The combined data on in vitro anti-trichinellosis activity, molecular structure investigations, and in silico prediction of drug-likeness outline the new 4-heteryl-2-amino-3,4-dihydro[1,3,5]triazino[1,2-*a*]benzimidazoles as potential drug candidates for the treatment of parasitic infections and motivate further research on their in vivo effectiveness.

## Data Availability

The data are available within the article.

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
