# Peer review of "Fused Triazinobenzimidazoles Bearing Heterocyclic Moiety: Synthesis, Structure Investigations, and In Silico and In Vitro Biological Activity"

_molecules, 2023, doi:10.3390/molecules28135034_

Round 1

Reviewer 1 Report

The article presented to me for review describes fused triazinobenzimidazoles containing a heterocyclic ring, which are highly interesting molecules. The authors have provided a motivation for the research on compounds that can be used as anti-malaria drugs in the introduction. The literature review on this topic is relatively modest but appears to be sufficient.

The authors have included a comprehensive spectral characterization of compounds 3a-f, as well as their biological activity. The procedures are clearly described and leave no room for doubt.

Studies could be conducted to investigate the reaction mechanism. DFT calculations should be extended to include these issues.

Only minor editorial corrections are required: (a typographical error in line 253: 96,47, in some places in the text, subscripts and superscripts are missing, in Supplementary fig.S6a and b are the same, fig. S1 – “delta” in 3d-table)

Author Response

  1. The DFT calculations were extended in order to investigate the posible reaction mechanism as Reviewer 1 recommended.
  2. A minor editorial corrections were made according Reviewer 1 recommendation - a typographical error in line 253: 96,47, in some places in the text, subscripts and superscripts are missing, in Supplementary fig.S6a and b are the same, fig. S1 – “delta” in 3d-table

Reviewer 2 Report

The authors described the synthesis, structure investigations, in silico and in vitro biological activity of fused triazinobenzimidazoles. I believe that the manuscript fulfills all the necessary criteria for publication in "Molecules" with minor revisions. I suggest providing experimental evidence, such as 2D NMR and/or X-ray crystallography, to support the assignment of the 3,4-dihydro form. Furthermore, I have included additional suggestions and comments:  

(1) See the abstract section. It appears to be quite general. It is crucial to include the key values related to the biological activity and the results of DFT calculations.

(2) See introduction. The paragraphs should be seamlessly connected without utilizing paragraph breaks.

(3) See Scheme 1. The catalyst loading of piperidine should be included (xx mol%). Additionally, the range of yield and reaction  time should be provided.

(4) See lines 101-106. From an organic perspective, it is important to explain, at the very least, the type of reactions involved in forming the 1,3,5-triazine ring. Additionally, organic chemists would find it intriguing to see a plausible mechanism and understand the role of the organocatalyst (piperidine).  

(5) See line 115. The letter "a" in [1,2-a] should be in italics. I recommend reviewing the entire manuscript.

(6) See Scheme 1. I strongly suggest including a Table that presents the type of substituent and corresponding yield. Also ¿Is the reaction time consistent across all products?

(7) See lines 117-124. In order to confirm the 3,4-dihydro form, X-ray crystallography is necessary. Alternatively, you can support it by referring to previous reports in the literature, which would provide a simple solution. Also, an explanation of 2D NMR spectra can support it.  

(8) See table 1. It is important to include the standard deviation for each measurement. Could you please clarify whether you used duplicates or triplicates?

(9) See 2.6. Prediction of the physico-chemical properties and drug-likeness. If you have conducted ADMET studies, I highly recommend complementing them with docking studies using the compound exhibiting the highest activity.

(10) See 3.2.1. General procedure for the synthesis of compound 3a-f. When reporting the 1H NMR data, ensure that the coupling constant (J) is presented with only one decimal place and in italics. Additionally, for each Rf value, specify the eluent used. I strongly recommend including MS data for all newly synthesized compounds.

(11) The conclusion section is excessively long. It needs to be shorten.

(12) See supplementary material. I strongly recommend including the MS spectra for all newly synthesized compounds.

Author Response

  1. As Reviewer 2  recommended 2D NMR of compounds 3b and 3c were added as examples to support the assignment of the 3,4-dihydro form.
  2. The abstract section was revised according Reviewer 2 recommendation in order to include the key values related to the biological activity and the results of DFT calculations.
  3. In introduction the paragraphs were connected without utilizing paragraph breaks as Reviewer 2 recommended.
  4. As Reviewer 2 recommended the type of reactions involved in formation of the 1,3,5-triazine ring and additionally the role of the organocatalyst (piperidine) was explained according a new Scheme 2 and the proposed mechanism was supported with DFT analysis.
  5. The letter "a" in [1,2-a] was write in italics as Reviewer 2 recommended.
  6. As Reviewer 2 recommended in the revised version of the manuscript were added the corresponding yield and reaction time for all synthesized compounds.(Section 3.2. Synthesis)
  7. As Reviewer 2 recommended in the revised manuscript was included the standard deviation for each measurement.
  8. At the moment the target of the synthesized compound was not unknown and that is why a docking studies were not performed.
  9. The General procedure for the synthesis of compound 3a-f was revised according Reviewer 2 recommendation - the coupling constant (J) were presented only with one decimal place and in italics; the eluent (benzene-methanol mixture 3:1, v/v) used in Rf value determination was specify in Section 3.1 General procedure; MS data for all newly synthesized compounds were added.
  10. The conclusion section in revised manuscript was shorten as Reviewer 2 recommended.
  11. The MS spectra of all new compounds were included as supplementary material.